# Malicious Traffic Detection Method for Power Monitoring Systems Based on Multi-Model Fusion Stacking Ensemble Learning

**DOI:** 10.3390/s25082614

**Published:** 2025-04-20

**Authors:** Hao Zhang, Ye Liang, Yuanzhuo Li, Sihan Wang, Huimin Gong, Junkai Zhai, Hua Zhang

**Affiliations:** 1State Grid Jibei Electric Power Co., Ltd., Beijing 100054, China; zhang_hao_1980@126.com (H.Z.); li_yuan_zhuo_24@163.com (Y.L.); 2Beijing Kedong Electric Power Control System Co., Ltd., Beijing 100192, China; liangye80@163.com (Y.L.); 17640260088@163.com (S.W.); 3State Key Laboratory of Networking and Switching Technology, Beijing University of Posts and Telecommunications, Beijing 100876, China; zjk20202020@126.com

**Keywords:** malicious traffic detection, multi-model fusion, power monitoring system, stacking strategy, machine learning

## Abstract

With the rapid development of the internet, the increasing amount of malicious traffic poses a significant challenge to the network security of critical infrastructures, including power monitoring systems. As the core part of the power grid operation, the network security of power monitoring systems directly affects the stability of the power system and the safety of electricity supply. Nowadays, network attacks are complex and diverse, and traditional rule-based detection methods are no longer adequate. With the advancement of machine learning technologies, researchers have introduced them into the field of traffic detection to address this issue. Current malicious traffic detection methods mostly rely on single machine learning models, which face problems such as poor generalization, low detection accuracy, and instability. To solve these issues, this paper proposes a malicious traffic detection method based on multi-model fusion, using the stacking strategy to integrate models. Compared to single models, stacking enhances the model’s generalization and stability, improving detection accuracy. Experimental results show that the accuracy of the stacking model on the NSL-KDD test set is 96.5%, with an F1 score of 96.6% and a false-positive rate of 1.8%, demonstrating a significant improvement over single models and validating the advantages of multi-model fusion in malicious traffic detection.

## 1. Introduction

In recent years, the rapid development of information technology has profoundly transformed all aspects of society, with the internet becoming an indispensable infrastructure for industry, business, and daily life. However, with the expansion and increased complexity of the internet, network attack methods have become more diverse and frequent, making network threats more severe. Among these threats, the surge of malicious traffic has become a significant challenge in the field of network security. Malicious traffic may carry harmful programs such as viruses and trojans, which may damage computer systems and data security and also be used for illegal activities such as launching distributed denial of service (DDoS) attacks, stealing sensitive information, etc., posing threats to personal privacy, corporate assets, and even national security.

Currently, methods for detecting malicious traffic are mainly divided into signature-based detection and anomaly-based detection [1]. Signature-based methods [2] rely on predefined rule libraries to match traffic characteristics (such as port numbers, protocol headers, and payload contents) with known attack signatures. While these methods have high detection accuracy for known threats, they require manual rule updates and struggle to cope with rapidly evolving unknown threats. In contrast, anomaly-based detection methods analyze user behavior patterns and identify abnormal traffic that deviates from normal behavior [3], effectively handling unknown threats. With the rapid development of artificial intelligence technologies, anomaly detection methods increasingly adopt machine learning algorithms [4], which autonomously learn complex traffic features to adapt to the diversity of malicious traffic. However, due to the blurred line between normal and malicious behavior, these methods tend to produce high false-positive rates. Furthermore, most current detection methods rely on single models, which often exhibit insufficient generalization and lower accuracy when addressing the diversity and complexity of malicious traffic.

To solve these issues, multi-model fusion methods have attracted increasing attention in recent years. Unlike single models, multi-model fusion combines the predictions of multiple machine learning models, improving the accuracy and reliability of predictions. This method effectively leverages the advantages of different models and better adapts to the complex and dynamic network traffic environment. Therefore, this study proposes a malicious traffic detection method based on multi-model fusion, aiming to address the limitations of single models. By utilizing the complementarity of models, it seeks to achieve higher detection accuracy, lower false-positive rates, and stronger generalization ability, providing more robust technical support for network security.

## 2. Background

### 2.1. Application of Traffic Detection in Power Monitoring Systems

With the continuous development of smart grids and power monitoring systems, the security of power networks has become an increasingly prominent issue, particularly regarding network attacks and malicious traffic threats. Power monitoring systems monitor the status and operation of power equipment through real-time data streams, typically relying on the integration of information technology (IT) and operational technology (OT). However, as network attack methods continue to evolve, power monitoring systems face multiple potential security threats from both internal and external sources, such as distributed denial of service (DDoS) attacks, data tampering, and intrusion into device control systems. These attacks can not only lead to equipment damage and data loss but also disrupt power supply, affecting the stability and security of the power grid.

To address these challenges, the application of malicious traffic detection methods in power monitoring systems is particularly important. Traffic detection technology enables real-time monitoring of the network traffic generated by power monitoring systems, allowing for the timely identification of abnormal traffic patterns and responses. Unlike traditional rule-based detection methods, machine learning- and ensemble learning-based traffic detection methods can automatically learn traffic patterns from historical data to identify unknown attacks and abnormal behaviors. This enables power monitoring systems to not only cope with known attacks but also to detect new types of attacks and unknown threats.

### 2.2. Overview of Malicious Traffic Detection Techniques

Malicious traffic detection is an important research area in network security, with common detection methods including signature-based, machine learning-based, and ensemble learning-based approaches. Traditional rule-based detection methods typically rely on predefined feature libraries or signatures to match malicious traffic, but this method can only identify known attacks and is weak in recognizing new attacks. In contrast, machine learning-based methods learn from labeled data to automatically extract features from traffic and identify unknown malicious activities. For example, Halimaa et al. [5] proposed an intrusion detection system based on SVM and naïve Bayes evaluated on the NSL-KDD dataset, with SVM outperforming naïve Bayes. Arijit C et al. [6] used k-means clustering and the SMO model, achieving 99.33% accuracy on the KDD CUP 99 dataset. Additionally, Roshan P et al. [7] used SVM and naïve Bayes models, achieving high accuracy (92–93%), precision (95–96%), and recall (98–99%) on a custom dataset. Despite progress in machine learning-based methods, they often rely on large amounts of labeled data and may face performance bottlenecks when dealing with complex contexts or new attacks. To improve detection performance, recent studies have started exploring ensemble learning methods, which combine multiple models to enhance detection accuracy and robustness. Simone A [8] proposed a malicious traffic detection model based on voting ensemble combining models like extreme learning machine and deep neural networks, achieving 92.5% accuracy on the NSL-KDD dataset. Raihan M et al. [9] proposed a voting-based ensemble detection model combining decision trees, k-nearest neighbors, and logistic regression, reaching 96.4% accuracy on the NSL-KDD dataset and significantly reducing false positives. Ensemble learning methods, compared to single models, can effectively improve the ability to recognize new attacks and further enhance the robustness and adaptability of traffic detection.

### 2.3. Overview of Ensemble Learning Methods

To significantly improve the generalization performance of learners, it is necessary to overcome common challenges of single learners, such as underfitting and overfitting [10]. Ensemble learning is an effective strategy that trains multiple independent learners and combines them using fusion strategies to form a more powerful learner. This method not only effectively reduces the bias that may arise from individual learners but also achieves precise learning and prediction for complex data through the complementarity of different learners. Common ensemble learning methods include voting, bagging, boosting, and stacking, with the stacking method being the primary focus of this paper. Unlike bagging and boosting, which rely on homogeneous models or sequential training, stacking integrates heterogeneous models through a meta-learner, enhancing both computational efficiency and robustness against overfitting [11].

The stacking method involves training multiple base learners in parallel and using a meta-learner to combine their predictions. Stacking adopts a multi-layer learning model where base learners make predictions on data, and their predictions are then used as new features for training the meta-learner, which makes the final prediction. Although stacking can theoretically combine the advantages of different learners, it may suffer from overfitting because predictions are generated using auxiliary training sets. To address this, k-fold cross-validation is often used to reduce the risk of overfitting. The literature [11] has proposed various stacking ensemble models that have performed well in multiple tasks.

Recent advances in cybersecurity detection frameworks have increasingly adopted stacking ensemble methods to address the limitations of single-model approaches. For instance, Cengiz and Gök demonstrated that stacking integrates gradient boosting and deep belief networks to achieve 98.7% accuracy in phishing detection, outperforming standalone models [12]. Similarly, our work aligns with the hierarchical feature fusion paradigm proposed in auction-based optimization frameworks, where base learners specialize in distinct data characteristics (e.g., temporal patterns, categorical features). This design ensures robustness against evasion attacks by leveraging complementary model strengths, as evidenced by recent benchmarks on NSL-KDD datasets.

### 2.4. Overview of Relevant Model Principles

This section briefly introduces the principles of several machine learning models selected for this study and demonstrates their application needs in malicious traffic detection and related fields.

Extreme gradient boosting (XGBoost) [13] is a member of the boosting family, utilizing ensemble learning to iteratively train multiple decision trees and combine their predictions with weighted results, thus improving the accuracy and efficiency of the model. The core optimization goal is:$$L(\theta )=\sum_{i=1}^{n}l\left(y_{i},{\hat{y}}_{i}\right)+\sum_{k=1}^{K}\Omega \left(f_{k}\right),$$

By introducing regularization terms (such as shrinkage and column sampling) to avoid overfitting, the regularization term controls the model’s complexity and reduces variance during training. In practical applications, XGBoost can handle custom loss functions to adapt to various prediction tasks. At the same time, XGBoost supports parallel training, significantly accelerating model training speed. Due to its efficiency, XGBoost is widely used in large-scale data prediction tasks.

Random forest (RF) [14] is a bagging method in ensemble learning that improves model accuracy and generalization by constructing multiple decision trees and combining their prediction results. Random forest increases the model’s diversity by randomly sampling the training data with replacements and randomly selecting features when splitting decision tree nodes, thus reducing the risk of overfitting. Finally, through voting or averaging, the predictions of each decision tree are combined to obtain the final prediction result:$$\hat{y}=\frac{1}{T}\sum_{t=1}^{T}f_{t}(x),$$

Due to its strong generalization ability, random forest performs excellently in classification and regression tasks, and is especially suitable for complex datasets.

Decision tree (DT) [15] is a classification and regression model based on a tree structure, where each leaf node represents a category and each internal node represents a decision condition. Decision trees recursively partition the data, progressively splitting it into subsets until stopping criteria are met. Common decision tree algorithms include ID3, C4.5, and CART, which differ in their feature selection methods. ID3 selects the optimal feature based on information gain, C4.5 improves ID3 by addressing continuous feature issues, and CART uses the Gini index for splitting feature selection. Taking the CART algorithm as an example, the Gini index is defined as:$$Gini(D)=1-\sum_{k=1}^{K}p_{k}^{2},$$

Decision trees are intuitive and easy to interpret, but they may suffer from overfitting, particularly when there are many features.

Logistic regression (LR) [16] is a linear model used for classification tasks, particularly binary classification. The model maps the results of linear regression to a range between 0 and 1 using a logistic function, representing the probability that a sample belongs to a certain category:$$P(y=1|x)=\sigma \left(w^{T}x+b\right)=\frac{1}{1+e^{-\left(w^{T}x+b\right)}},$$

The optimization goal of the model is to maximize the log-likelihood function:$$\underset{w}{\max}\sum_{i=1}^{n}\left[y_{i}\log{\hat{y}}_{i}+\left(1-y_{i}\right)\log\left(1-{\hat{y}}_{i}\right)\right],$$

The key advantage of logistic regression is that it not only predicts the category but also outputs the probability of each category, which is particularly important for decision tasks. Unlike generative models, logistic regression does not require assumptions about the data distribution, avoiding potential issues with inaccurate distribution assumptions. Additionally, logistic regression’s optimization process is simple and suited for handling linearly separable datasets.

Naïve Bayes (NB) [17] is a probabilistic classification algorithm that assumes the independence of the data’s features. Although this assumption may not hold in many real-world applications, naïve Bayes excels in many practical problems, such as text classification and spam detection. Based on Bayes’s theorem, naïve Bayes classifies by calculating posterior probabilities:$$P(y|x)=\frac{P(x|y)P(y)}{P(x)}\propto P(y)\prod_{i=1}^{n}P\left(x_{i}|y\right),$$

Common variants include Gaussian naïve Bayes (for continuous features), multinomial naïve Bayes (for count data), and Bernoulli naïve Bayes (for binary data). It has low computational complexity, making it suitable for large-scale datasets and simple implementation.

LightGBM (light gradient boosting machine) [18] is an efficient machine learning algorithm based on the gradient boosting framework. It employs a leaf-wise growth strategy that splits leaves based on the highest gradient, speeding up training compared to traditional depth-first growth strategies. LightGBM also reduces memory consumption by discretizing continuous features using histogram-based algorithms and improves precision and efficiency by employing gradient-based one-side sampling (GOSS). LightGBM is suitable for large datasets, supports distributed training and GPU acceleration, and offers advantages in training time and memory usage compared to XGBoost, though XGBoost has better model interpretability and regularization.

Adaptive boosting (AdaBoost) [19] is an iterative ensemble learning method that combines multiple weak classifiers to create a strong classifier. In each iteration, AdaBoost adjusts the weights of each sample based on prediction errors, focusing the weak classifiers on harder-to-classify samples. Ultimately, AdaBoost combines the predictions of these classifiers with weighted averages to form a strong classifier. The optimization goal is:$$F(x)=\sum_{t=1}^{T}\alpha_{t}h_{t}(x),$$

AdaBoost excels in preventing overfitting and providing high classification accuracy. One advantage of AdaBoost is its flexibility in choosing the weak classifiers according to the task’s needs, such as decision trees or logistic regression.

## 3. Malicious Traffic Detection Method Based on Stacking Ensemble Learning

This section investigates a malicious traffic detection method based on multi-model fusion, aiming to improve the model’s ability to recognize malicious traffic patterns in complex network environments and addressing the limitations of a single model in generalization and accuracy. To achieve this, a detection framework was designed using the stacking ensemble strategy, which integrates the predictions of multiple base learners to enhance the recognition precision of malicious traffic patterns. To further optimize feature expression, we performed feature selection on the NSL-KDD dataset and used the random forest (RF) algorithm to identify key features.

Compared to existing research, using RF for feature selection has clear advantages. RF is an ensemble algorithm based on decision trees, capable of automatically evaluating the importance of each feature and selecting those most critical for improving model performance. In contrast to traditional feature selection methods (such as univariate selection based on statistical tests), RF can handle nonlinear relationships between features and effectively avoid overfitting. Moreover, RF is particularly efficient in processing high-dimensional data. By using random sampling and feature subset selection, RF reduces dependency on the full dataset, thus accelerating training speed and enhancing model robustness. These advantages allow RF to significantly improve model performance in complex network environments and increase the accuracy of malicious traffic detection, distinguishing it from traditional methods.

### 3.1. Malicious Traffic Detection Framework

This method utilizes the NSL-KDD dataset to build the model, which is widely used in traffic detection, serving as a benchmark dataset for many network security traffic detection experiments [20]. Although the NSL-KDD dataset is not specifically designed for power monitoring systems, its attack categories (e.g., DoS, R2L) align with prevalent threats in power grid cybersecurity, such as unauthorized device access and denial-of-service attacks targeting SCADA systems. While newer datasets like CIC IDS 2023 exist, NSL-KDD remains a widely accepted benchmark for evaluating model generalizability in anomaly detection tasks. Based on the NSL-KDD dataset, the workflow of the malicious traffic detection framework in this paper is shown in Figure 1. The components of this framework include the following.

Data Preprocessing: This involves digitization, normalization, and categorization. Categorical features are converted into numerical features using one-hot encoding, Min–max normalization is applied to the data, and the data are classified according to traffic categories.

Feature Selection: This module uses a random forest classifier to select the 20 most important features from the 41 features of the NSL-KDD dataset.

Malicious Traffic Classifier: The stacking ensemble learning method is used as the model fusion strategy to identify whether the input key feature data represent malicious network traffic.

### 3.2. Data Preprocessing and Feature Selection

The NSL-KDD dataset consists of 41 continuous or discrete features, as shown in Table 1. The traffic in the dataset can be divided into five categories: DoS (denial of service), probe, R2L (remote to local), U2R (user to root), and normal. Some attack types in the test set do not appear in the training set, which better reflects the model’s actual malicious traffic detection capability. Table 2 presents the number of traffic records in both the training and test sets, along with their distribution based on network traffic type (normal or attack type). The training dataset contains 125,973 records, and the test dataset contains 22,544 records.

The features in the dataset can be categorized into four types: basic features, content features, time-based network traffic statistical features, and host-based network traffic statistical features.

Basic Features (1–9): These features can be obtained from the header of the packet without needing to inspect the payload. They contain basic information about the packet, such as duration, protocol type, etc.

Content Features (10–22): These features provide information about the raw data packet. For attacks like U2R and R2L, since they do not exhibit frequent sequence patterns in data records like DoS attacks, but are embedded within the packet payload, distinguishing a single data packet from normal connections is difficult. These features provide access to the payload data.

Time-Based Network Traffic Statistical Features (23–31): These features help identify relationships between the current connection and previous ones within a time window of 2 s, reflecting connections more accurately. They include statistics such as how many attempts were made to establish a connection with the same host. These are primarily counts and ratios not related to the content of traffic input.

Host-Based Network Traffic Statistical Features (32–41): While time-based features focus on the most recent 2 s window, some probing attacks use slow attack patterns to scan hosts or ports, which may occur at frequencies exceeding the 2 s threshold. Therefore, these features are classified based on the target host, using a time window of 100 connections to capture the statistical information of the current connection relative to the previous 100 connections. They aim to mark attacks that span over a 2 s window.

#### 3.2.1. Data Preprocessing

Data preprocessing is a data mining technique that transforms raw collected data into a useful and effective format [21]. It is a critical step in every machine learning project because the dataset used significantly impacts the performance of the built system. The purpose of data preprocessing is to use only the key data that contributes to building an efficient system [22].

The data preprocessing steps in this study mainly include numerical transformation, normalization, and classification, as shown in Figure 2.

The NSL-KDD dataset includes both numerical and categorical features. To enable the model to effectively learn and predict, categorical features must be digitized. One-hot encoding is used to convert categorical features into numerical features. The dataset contains three categorical features: “protocol_type,” “service,” and “flag.” The “protocol_type” feature has three attributes: “TCP,” “UDP,” and “ICMP.” These are one-hot encoded into three vectors: (1,0,0), (0,1,0), and (0,0,1). Similarly, the “service” feature has 70 attributes, and the “flag” feature has 11 attributes that also need to be one-hot encoded.

The NSL-KDD dataset is an improved version of the KDD CUP 99 dataset. Compared to the original KDD dataset, the NSL-KDD dataset removes redundant records from the training set and does not contain duplicate records in the test set. Therefore, normalization is directly applied here. The min–max normalization method is used for data transformation, mapping the result values to the range [0, 1]. The transformation function is given by Equation (1):(1)$$x^{\prime }=\frac{x-\min(x)}{\max(x)-\min(x)}$$
where *x*′ is the transformed data, min(*x*) is the minimum value in the dataset, and max(*x*) is the maximum value in the dataset.

Classification involves categorizing the available data into distinct classes (DoS, Probe, U2R, R2L, and normal). The specific attack types are shown in Table 3. After classification, the categorical attributes are extracted and one-hot encoded.

#### 3.2.2. Feature Selection

From the perspective of feature engineering, the NSL-KDD dataset has already completed several key steps in feature engineering, including feature availability, feature collection, and the definition and calculation of derived features. Therefore, feature selection can begin directly when using this dataset for detection experiments.

Feature selection refers to the process of selecting a subset of features from the dataset that contribute significantly to the task objective, with the goal of simplifying the data, reducing the feature dimension, and decreasing the computational overhead of the model. Feature selection helps eliminate redundant or irrelevant features, thus improving the model’s generalization ability. Research shows that the random forest (RF) algorithm performs well in feature selection [23]. Therefore, this study uses the random forest algorithm for feature selection as the basis for subsequent experiments.

First, we construct a random forest model consisting of multiple decision trees using the training dataset. In this study, the Gini index method is mainly used to compute feature importance: during the decision tree splitting process, a split based on a particular feature leads to a change in the Gini index (i.e., an improvement in data purity). Random forest can calculate the total contribution of a specific feature to the Gini index at the splitting nodes across all trees, which is used as the importance score for that feature. The calculation method is shown in Equation (2):(2)$$I(f)=\sum_{t=1}^{T}\Delta_{Gini}(f_{j},t)$$
where $I(f_{j})$ is the importance of feature $f_{j}$, *T* is the total number of decision trees, and $\Delta_{Gini}(f_{j},t)$ represents the improvement in the Gini index brought by feature $f_{j}$ at the *t*-th tree.

Finally, features are ranked according to their importance scores, and the highest-scoring features are selected as the final input features. Figure 3 shows the importance levels of all features in the dataset determined by the random forest algorithm.

Based on the feature importance levels, the 20 features with the highest importance are selected from the dataset. The relevant information about these features is shown in Table 4.

### 3.3. Stacking Multi-Model Fusion Strategy

The stacking method aggregates weak classifiers to form a strong classification model, solving the problem of malicious traffic detection in complex internet environments. Given the complexity and performance costs of malicious traffic detection, the stacking method proposed in this work consists of two layers: the first layer and the second layer. In the stacking fusion method, the complete training dataset is used to train the first layer of learners. The predictions generated by the first-layer learners are then treated as new features and serve as inputs to the second-layer meta-learner. Thus, stacking is a parallel approach. The first layer includes three heterogeneous weak learners, while the second layer employs a logistic regression (LR) model as the meta-learner, which combines the results from the three weak learners to generate the final prediction. Additionally, to prevent overfitting, k-fold cross-validation is applied during the training of the first-layer learners, with fivefold cross-validation being specifically chosen in this study. A random 20% of the dataset is selected as the test set.

The process of using the stacking method to detect malicious traffic is as follows.

Phase 1: In the stacking method, assume there are *L* weak learners *M*_1_, *M*_2_, …, *M_L_* (in this case, *L* = 3). Each weak learner is trained using the training set *D*_train_ and generates predictions for both the training set *D*_train_ and the test set *D*_test_:

For the training set, the predictions generated by the weak learners form a new training dataset *D’* = [*D*_1_, *D*_2_, …, *D_L_*], where *D_i_* represents the predictions from the *i*-th weak learner for *D*_train_.

For the test set, the predictions generated by the weak learners form a new test dataset *T’* = [*T*_1_, *T*_2_, …, *T_L_*], where *T_i_* represents the predictions from the *i*-th weak learner for *D*_test_.

Phase 2: The second-layer meta-learner uses a logistic regression model, with *D’* (the feature dataset generated by the first-layer learners) as its input. *D’* serves as the training data for the meta-learner, while the true labels (i.e., the original class labels) are retained to guide the supervised learning of the meta-learner. The logistic regression model learns the relationship between the predictions of the weak learners and the true labels in *D’* by optimizing the loss function of logistic regression. This allows the meta-learner to determine the weights of each weak learner’s predictions in the final output.

After training, the meta-learner predicts the feature data *T’* from the test set and produces the final combined classification result.

Based on the process described above, the algorithm design of the multi-model fusion detection model based on the stacking strategy is shown in Algorithm 1.
**Algorithm 1:** Multi-Model Fusion Detection Model Based on Stacking Strategy       Input: Training set $D=\left\{x_{i},y_{i}\right\}$, i∈{1,2…,n}       First-layer learners: *M*_1_, *M*_2_, …, *M_L_*       Meta-learner: *M*       Step 1: Train the first-layer learners on the complete training dataset *D*. For each first-layer learner *M_l_* (*l* = 1, 2, 3, …, *L*), train on the complete training dataset *D* to obtain learning model $h_{l}$.       Step 2: Construct a new dataset. For each sample $\left\{x_{i},y_{i}\right\}$i∈{1,2…,n}, use the first-layer learners $h_{l}$ to predict *x_i_*, generating prediction values $d_{il}=h_{l}\left(x_{i}\right)$. All predictions from the first-layer learners $\left(d_{i1},d_{i2},\ldots ,d_{iL}\right)$ are combined as new features and combined with the original labels $y_{i}$ to construct a new training dataset $D^{\prime }$.       Step 3: Train the meta-learner *M* on the new training dataset $D^{\prime }$ to obtain the meta-learning model $h^{\prime }$.       Step 4: Final model prediction. The model output is: $H(x)=h^{\prime }\left(h_{1}(x),h_{2}(x),\ldots h_{L}(x)\right)$.       Output: Stacking ensemble model H(x).

The specific detection process of the stacking ensemble method with fivefold cross-validation is shown in Figure 4. As per the flowchart, the process of obtaining the training set *D_cv_* for the meta-learner *M* using cross-validation is analyzed. For example, given a preprocessed dataset *D*, it is randomly split into *K* equal folds {*D*_1_, *D*_2_, …, *D_K_*} (with *K* = 5 in this study), where $d_{k}^{tr}$ and $d_{k}^{te}$ serve as the training and testing sets for the *k*-th fold in *K*-fold cross-validation. In the first layer, different machine learning models (*M*_1_, *M*_2_, …, *M_L_*) are given (in this study, *L* = 3), and each *M_L_* model is trained on the training set $d_{k}^{tr}$ and makes predictions on the test set $d_{k}^{te}$ instances. The prediction results of all first-layer learners are combined to form a new dataset *D_cv_*, which is then used as the input for the meta-learner *M*. The goal of the meta-learner *M* is to learn how to combine the predictions of the first-layer learners to perform the classification task more accurately. At the end of the entire cross-validation process for each model *M_L_*, the data (*D_cv_*) are the collection of results from the *L* models.

## 4. Stacking Ensemble Model Testing

In this experiment, the proposed stacking ensemble model was tested. Through experiments with single models and different model combinations, the final ensemble model was derived. XGBoost, RF, and LightGBM were used as the first-layer learners in the stacking model, and LR was used as the meta-learner. The experimental results show that the proposed ensemble model performs well in malicious traffic detection, achieving an accuracy of 96.5%, precision of 97.1%, recall of 96.2%, F1 score of 96.6%, and a false-positive rate of 1.8%. This demonstrates that the model not only has high accuracy but also a low false-positive rate.

### 4.1. Experimental Setup

The experiment was conducted in a Python 3.7.0 development environment based on the PyTorch framework combined with machine learning libraries such as Scikit-learn 0.22.1 and XGBoost 1.0.2. The experimental environment used a laptop with the Windows 10 operating system, 16 GB of memory, an Intel i7-10510U CPU (1.80 GHz), and an MX 350 GPU as the accelerator.

The dataset used in the experiment was the NSL-KDD dataset. In the NSL-KDD dataset, the flaws in the original KDD CUP 99 dataset [24] have been removed. Although there are still issues in the NSL-KDD dataset [25], they do not affect its applicability or the validity of the research results in this study. Each record in the NSL-KDD dataset consists of network traffic data with 41 defined attributes (e.g., protocol type, service, and flags), which are labeled as either normal or attack (probe, DoS, U2R, and R2L). The 41 features of the NSL-KDD dataset are referenced in Table 1, the distribution of network traffic types (normal or attack) is shown in Table 2, and the attack types are referenced in Table 3.

### 4.2. Experimental Results and Analysis

In this study, the following models were selected as candidate first-layer learners for the stacking model: XGBoost, random forest (RF), decision tree (DT), naïve Bayes (NB), LightGBM, and AdaBoost. LR was used as the meta-learner, and fivefold cross-validation was applied in the experiment. In the experimental phase, accuracy and F1 score were used as the criteria for selecting the best learners to find the optimal combination of these models.

The first experiment was conducted on each first-layer learner based on the NSL-KDD dataset, and the detection performance of each model is shown in Table 5. In Table 5, it can be seen that XGBoost, RF, and LightGBM have the best classification performance, with accuracies of 92.3%, 91.6%, and 90.4% and F1 scores of 92.4%, 91.6%, and 90.5%, respectively. The classification performance of NB is the worst, with an accuracy of 85.7% and an F1 score of 85.9%.

Selecting the best model combination based solely on the performance of individual learners is not sufficiently accurate. To determine the best stacking ensemble model, experiments were conducted to train and evaluate different combinations of first-layer learners. The experimental results of the top five combinations are shown in Table 6.

From the experimental results, the ensemble model with XGBoost, RF, and LightGBM as the first-layer learners has the best detection performance, with an accuracy of 96.5% and an F1 score of 96.6%. In the top five combinations, XGBoost and RF appear most frequently, each appearing in four combinations. LightGBM and AdaBoost appear three times each, NB appears once, and DT does not appear in any of the top-five combinations. The performance of the stacking ensemble model is related to the independence of the learners at the same level. XGBoost, RF, LightGBM, and AdaBoost are all decision tree-based algorithms, so combinations containing decision trees tend to perform worse.

Additionally, the ensemble model combining RF, NB, and XGBoost achieved an accuracy and F1 score of 95.4%, both higher than their individual model performances. This shows that although some first-layer learners performed poorly in earlier experiments, their recognition ability can be effectively improved when combined with other learners, suggesting that the stacking ensemble method can enhance the model’s performance in detecting malicious traffic.

Based on the experimental results, this study finally selected XGBoost, RF, and LightGBM as the first-layer learners for the stacking model and LR as the meta-learner. The detailed experimental results of this model are shown in Table 7. The results show that the stacking ensemble model has an accuracy of 96.5%, precision of 97.1%, recall of 96.2%, F1 score of 96.6%, and a false-positive rate of only 1.8%, achieving excellent detection performance. These results demonstrate that the stacking ensemble method significantly improves the detection accuracy compared to single models while maintaining a low false-positive rate. We have analyzed these base learners below.

XGBoost: While excelling in capturing temporal patterns in connection logs (e.g., repeated failed login attempts), XGBoost’s reliance on gradient-based feature importance ranking led to biased predictions for rare attacks like R2L (recall of 89.3% vs. Stacking’s 96.2%).

Random Forest: Its ensemble of decision trees exhibited high variance in feature selection, causing inconsistent performance across attack categories (e.g., 90.1% F1 score for DoS vs. 85.4% for U2R.

LightGBM: Although specialized in handling imbalanced datasets (e.g., 91.2% F1 score for U2R), its leaf-wise growth strategy resulted in overfitting-prone predictions for low-prevalence attacks (false-positive rate of 4.7%).

These limitations highlight the need for a hybrid approach to achieve balanced detection capabilities. The stacking framework addresses these gaps by synergizing complementary strengths: XGBoost’s temporal sensitivity, RF’s feature diversity, and LightGBM’s imbalance tolerance, while the meta-learner (LR) calibrates their predictions to reduce variance.

Regarding time performance, due to the layered framework in the ensemble learning method using LR as the meta-learner to process the prediction results from XGBoost, RF, and LightGBM, the overall time cost is slightly higher than that of a single algorithm. However, this trade-off is justified in our target deployment scenario. Specifically, power monitoring systems typically adopt a bypass architecture, where traffic data are passively captured and stored for offline analysis. In such settings, real-time responsiveness is not critical, as the primary objective is to achieve comprehensive threat coverage rather than millisecond-level detection speed. As shown in Table 8, the 517 s runtime includes preprocessing, feature engineering, and fivefold cross-validation, which are essential for ensuring model robustness.

In summary, the XGBoost algorithm uses feature-level column sampling and regularization strategies to reduce model complexity and the risk of overfitting. It pays more attention to the misclassified samples from the previous iteration, thus improving the classification accuracy of connection features in the dataset, especially for time- and host-based features. The random forest algorithm performs well in classifying basic and content features of connections in the dataset. Moreover, the random forest generation process builds unbiased estimates of error, reducing classification variance and improving detection accuracy. LightGBM uses gradient-based one-side sampling and exclusive feature bundling algorithms to reduce computation and the number of features, enhancing model efficiency. The leaf-wise strategy for tree construction further reduces bias by selecting the leaf node with the highest information gain at each level.

## 5. Conclusions

This paper proposes a malicious traffic detection method based on a stacking model. By combining XGBoost, RF, and LightGBM as first-layer learners and LR as the meta-learner, the classification performance of malicious traffic detection is significantly improved. Experimental results show that the model achieves an accuracy of 96.5%, an F1 score of 96.6%, precision of 97.1%, recall of 96.2%, and a false-positive rate of only 1.8% on the NSL-KDD dataset. Compared to single models, the stacking ensemble model effectively improves detection accuracy and maintains a low false-positive rate, validating the effectiveness of the stacking method.

Our approach aligns with the meta-learning paradigm proposed by Zhu et al. [26], where stacking’s hierarchical aggregation reduces reliance on individual model calibration. Their work on federated learning vulnerabilities further validates that stacking’s meta-learner can mitigate overfitting risks without manual hyperparameter adjustments, particularly in data-scarce environments like power grids.

In power monitoring systems, real-time processing and accuracy are core requirements for detection systems. Although the training and prediction time of the stacking model is longer, its high accuracy and low false-positive rate can significantly reduce false positives and false negatives in practice, minimizing potential losses caused by malicious traffic attacks. Especially in the complex environment of power data, this model can flexibly adapt to diverse attack patterns, enhancing the system’s overall defense capabilities. The malicious traffic detection method proposed in this study not only offers advantages in improving detection accuracy but also provides a more reliable security mechanism for power monitoring systems. Future research could further optimize the real-time performance of the model and explore more efficient fusion strategies to improve its practical application.

## Figures and Tables

**Figure 1 sensors-25-02614-f001:**
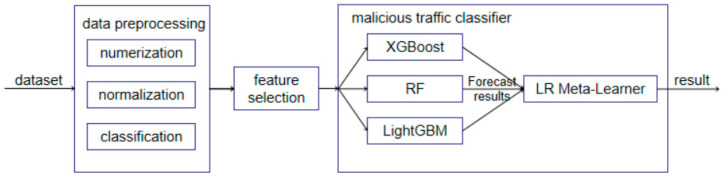
Malicious traffic detection framework.

**Figure 2 sensors-25-02614-f002:**
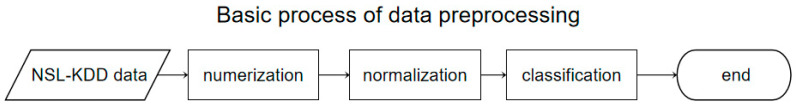
Data preprocessing steps.

**Figure 3 sensors-25-02614-f003:**
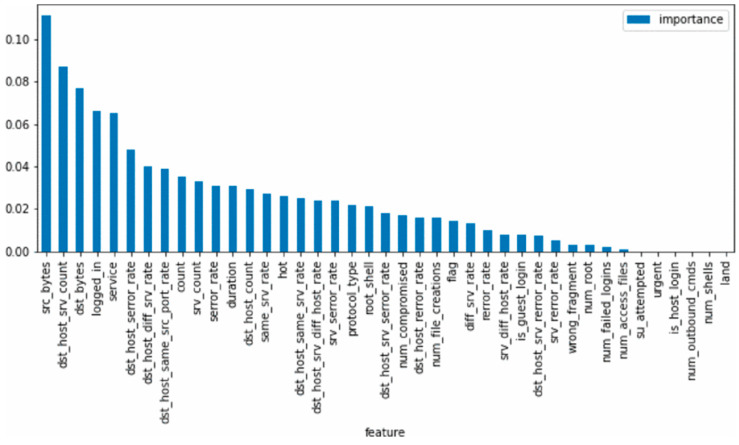
Feature importance levels.

**Figure 4 sensors-25-02614-f004:**
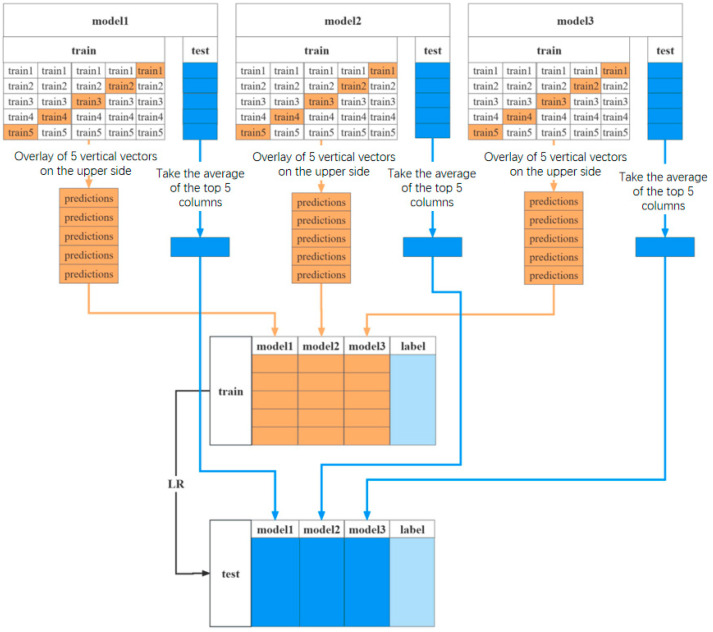
Detailed stacking fusion detection process.

**Table 1 sensors-25-02614-t001:** Features of the NSL-KDD dataset.

No.	Feature Name	No.	Feature Name
1	duration	22	is_guest_login
2	protocol_type	23	count
3	service	24	srv_count
4	flag	25	serror_rate
5	src_bytes	26	srv_serror_rate
6	dst_bytes	27	rerror_rate
7	land	28	srv_rerror_rate
8	wrong_fragment	29	same_srv_rate
9	urgent	30	diff_srv_rate
10	hot	31	srv_diff_host_rate
11	num_failed_logins	32	dst_host_count
12	logged_in	33	dst_host_srv_count
13	num_compromised	34	dst_host_same_srv_rate
14	root_shell	35	dst_host_diff_srv_rate
15	su_attempted	36	dst_host_same_src_port_rate
16	num_root	37	dst_host_srv_diff_host_rate
17	num_file_creations	38	dst_host_serror_rate
18	num_shells	39	dst_host_srv_serror_rate
19	num_access_files	40	dst_host_rerror_rate
20	num_outbound_cmds	41	dst_host_srv_rerror_rate
21	is_hot_login		

**Table 2 sensors-25-02614-t002:** Traffic distribution of the NSL-KDD dataset.

Dataset	Normal	DoS	Probe	U2R	R2L	Total
Training	67,343 (53%)	45,927 (37%)	11,656 (9.11%)	52 (0.04%)	995 (0.85%)	125,973
Testing	9711 (43%)	7458 (33%)	2421 (11%)	200 (0.92%)	2754 (12.1%)	22,544

**Table 3 sensors-25-02614-t003:** Attack types in the NSL-KDD dataset.

Attack Type	Specific Attacks
**DoS**	back, land, neptune, pod, smurf, teardrop, mailbomb, apache2, processtable, udpstorm
**Probe**	ipsweep, nmap, portsweep, satan, mscan, saint
**R2L**	ftp_write, guess_passwd, imap, multihop, phf, spy, warezclient, warezmaster, sendmail, named, snmpgetattack, snmpguess, xlock, xsnoop, worm
**U2R**	buffer_overflow, load-module, perl, rootkit, httptunnel, ps, sqlattack, xterm

**Table 4 sensors-25-02614-t004:** The 20 most important features from the NSL-KDD dataset.

No.	Feature	Description
1	src_bytes	Number of bytes sent from the source host to the destination host
2	dst_host_srv_count	Number of connections with the same destination host and service as the current one
3	dst_bytes	Number of bytes sent from the destination host to the source host
4	logged_in	1 if login is successful, otherwise 0
5	service	The network service type on the destination host
6	dst_host_serror_rate	Percentage of connections with SYN errors among those with the same destination host
7	dst_host_diff_srv_rate	Percentage of connections with different services among those with the same destination host
8	dst_host_same_src_port_rate	Percentage of connections with the same source port on the same destination host
9	count	Number of connections with the same destination host as the current one
10	srv_count	Number of connections with the same service as the current one
11	serror_rate	Percentage of connections with SYN errors among those with the same destination host
12	duration	Duration of the connection
13	dst_host_count	Number of connections with the same destination host
14	same_srv_rate	Percentage of connections with the same service as the current one at the same destination host
15	hot	Number of times sensitive system files or directories were accessed
16	dst_host_same_srv_rate	Percentage of connections with the same service at the same destination host
17	dst_host_srv_diff_host_rate	Percentage of connections from different source hosts with the same service at the same destination host
18	srv_serror_rate	Percentage of SYN error connections among those with the same service as the current one
19	protocol_type	Type of protocol used (e.g., TCP, UDP, ICMP)
20	root_shell	1 if superuser privileges were gained, otherwise 0

**Table 5 sensors-25-02614-t005:** Comparison of single-learner experimental results.

Algorithm	Accuracy	F1
XGBoost	92.3%	92.4%
RF	91.6%	91.6%
DT	87.7%	87.8%
NB	85.7%	85.9%
LightGBM	90.4%	90.5%
AdaBoost	89.1%	89.2%

**Table 6 sensors-25-02614-t006:** Experimental results of different combinations of stacking ensemble models.

First-Layer Learners	Accuracy	F1
XGBoost, RF, LightGBM	96.5%	96.6%
XGBoost, RF, AdaBoost	96.2%	96.3%
XGBoost, LightGBM, AdaBoost	96.0%	95.9%
RF, LightGBM, AdaBoost	95.7%	95.7%
RF, NB, XGBoost	95.4%	95.4%

**Table 7 sensors-25-02614-t007:** Experimental results of the stacking ensemble model.

Model	Accuracy	Precision	Recall	F1	False-Positive Rate
Proposed Model	96.5%	97.1%	96.2%	96.6%	1.8%

**Table 8 sensors-25-02614-t008:** Comparison of runtime for different models.

Model	XGBoost	RF	LightGBM	Proposed Model
Runtime, s	269	240	108	517

## Data Availability

No new data were created or analyzed in this study. Data sharing is not applicable to this article.

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
