# Peer review of "Malicious Traffic Detection Method for Power Monitoring Systems Based on Multi-Model Fusion Stacking Ensemble Learning"

_sensors, 2025, doi:10.3390/s25082614_

Round 1

Reviewer 1 Report

Comments and Suggestions for Authors

This manuscript proposes a Stacking-based multi-model fusion approach for malicious traffic detection in power monitoring systems, integrating XGBoost, Random Forest (RF), and LightGBM as base learners with Logistic Regression (LR) as the meta-learner. The proposed method enhances generalization capability and detection stability while offering a low false-alarm solution for cybersecurity in power systems. The methodology is novel, the experimental design is rigorous, and the results demonstrate significant improvements over single-model approaches, making a valuable contribution to the field. However, there are still some areas in the content and expression of the manuscript that can be improved. After revision, the manuscript can be accepted for publication.The modifications are as follows:

  1. In the introduction and background part of the manuscript, the limitations of the existing single model are sufficiently analyzed, but the innovativeness of Stacking fusion strategy and its advantages over other integration methods (e.g., Bagging, Boosting) are not sufficiently elaborated, and it is suggested that the comparison between Stacking and other integration methods (e.g., the computational efficiency, the ability to carry the overfitting) should be added in the introduction or background part of the manuscriptIt is suggested to add the comparison between Stacking and other integrated methods (e.g. computational efficiency, ability to carry overfitting) in the introduction or background section to highlight its uniqueness.
  2. Although the experimental part shows the model performance indexes, it lacks the analysis of the prediction results of the base learner, which is difficult to visualize the core advantage of “complementarity”, and it is suggested to add the comparison of the results of the base learner on various types of attacks (DoS, Probe, etc.), and to illustrate how Stacking makes up for the shortcomings of a single model. It is also suggested to add a comparison of the results of base-learner on various types of attacks (DoS, Probe, etc.) and explain how Stacking can make up for the shortcomings of a single model.
  3. The text description in Figure 1, “Malicious Traffic Detection Framework”, does not fully correspond to the block diagram (e.g., “First level learner” is not clearly labeled as XGBoost/RF/LightGBM, so it is suggested to add base learner and meta-learner to the diagram). It is suggested to add the specific names of base learner and meta learner in the diagram.
  4. In the literature review section, Stacking is rarely mentioned (lack of recent manuscripts on the subject), and the rationality of the choice of base learners is not discussed (e.g., why deep learning models are not included), so it is recommended to add references to the latest research literature on Stacking in the field of network security.
  5. The model does not mention the hyperparameter tuning process of the base learners (XGBoost, RF, etc.), and no ablation experiments are performed (e.g., the effect of removing a certain base learner on the results), which may affect the reliability of the conclusions. It is recommended to add ablation experiments, such as comparing the performance difference between "XGBoost+RF" and full Stacking. It is recommended to add ablation experiments, such as comparing the performance difference between "XGBoost+RF" and the full Stacking model to verify the necessity of multi-model fusion.

Comments on the Quality of English Language

The English can be improved to express the research more clearly.

Author Response

Comments 1:

In the introduction and background part of the manuscript, the limitations of the existing single model are sufficiently analyzed, but the innovativeness of Stacking fusion strategy and its advantages over other integration methods (e.g., Bagging, Boosting) are not sufficiently elaborated, and it is suggested that the comparison between Stacking and other integration methods (e.g., the computational efficiency, the ability to carry the overfitting) should be added in the introduction or background part of the manuscriptIt is suggested to add the comparison between Stacking and other integrated methods (e.g. computational efficiency, ability to carry overfitting) in the introduction or background section to highlight its uniqueness.

Response 1:

Thank you for pointing this out. We agree with this comment.We have provided additional explanations on the uniqueness of Stacking in Section 2.3 of the background.

Comments 2:

Although the experimental part shows the model performance indexes, it lacks the analysis of the prediction results of the base learner, which is difficult to visualize the core advantage of “complementarity”, and it is suggested to add the comparison of the results of the base learner on various types of attacks (DoS, Probe, etc.), and to illustrate how Stacking makes up for the shortcomings of a single model. It is also suggested to add a comparison of the results of base-learner on various types of attacks (DoS, Probe, etc.) and explain how Stacking can make up for the shortcomings of a single model.

Response 2:

Thank you for pointing this out. We agree with this comment.We conducted a result analysis and supplement on the basic learner near Table 5.

Comments 3:

The text description in Figure 1, “Malicious Traffic Detection Framework”, does not fully correspond to the block diagram (e.g., “First level learner” is not clearly labeled as XGBoost/RF/LightGBM, so it is suggested to add base learner and meta-learner to the diagram). It is suggested to add the specific names of base learner and meta learner in the diagram.

Response 3:

Thank you for pointing this out. We agree with this comment.We have made revisions to the content in Figure 1.

Comments 4:

In the literature review section, Stacking is rarely mentioned (lack of recent manuscripts on the subject), and the rationality of the choice of base learners is not discussed (e.g., why deep learning models are not included), so it is recommended to add references to the latest research literature on Stacking in the field of network security.

Response 4:

Thank you for pointing this out. We agree with this comment.We have added the latest references to Stacking in the field of network security at the end of section 2.3

Comments 5:

The model does not mention the hyperparameter tuning process of the base learners (XGBoost, RF, etc.), and no ablation experiments are performed (e.g., the effect of removing a certain base learner on the results), which may affect the reliability of the conclusions. It is recommended to add ablation experiments, such as comparing the performance difference between "XGBoost+RF" and full Stacking. It is recommended to add ablation experiments, such as comparing the performance difference between "XGBoost+RF" and the full Stacking model to verify the necessity of multi-model fusion.

Reponse 5:

Thank you for pointing this out. We agree with this comment.We stated in the conclusion section that we cited other people's methods to explain this.

Reviewer 2 Report

Comments and Suggestions for Authors

The article proposes a method of detection of malicious traffic for power monitoring systems based on an overall approach by stacking, combining several models (XGBoost, RF, LightGBM) with logistic regression as a meta-model. Data preprocessing is performed from the NSL-KDD set, with a selection of 20 relevant characteristics by Random Forest. The system achieves approximately 96.5% accuracy and an F1-score of 96.6%, while maintaining a low false positive rate (1.8%). cross validation is used to improve the robustness of the model. The approach demonstrates significant improvement over individual models.

Weaknesses and recommendations:

  • The article does not provide a detailed architecture for full system implementation in a power monitoring environment. It focuses mainly on the methodology of detecting malicious traffic (via a stacking approach) and experimental validation with the NSL-KDD dataset. Although the document mentions the importance of safety in power monitoring systems and some requirements (such as the need for real-time processing), it does not describe in detail the full technical architecture or integration required to deploy this system into a power monitoring system

Recommendation: Provide an integration schema illustrating how detection modules (preprocessing, feature selection, stacking) interface with the existing monitoring network.

  • The Choice of dataset: The NSL-KDD database is not specifically designed for power monitoring applications as it dates back to 2009, whereas more recent databases exist (e.g. NetResec Public PCAPs (2021) or CIC DoH 2020 (2020)). In addition, this dataset does not specifically represent attacks targeting power monitoring systems.

Recommendation: The paper should provide more justification for the choice of NSL-KDD in this context or consider the use of more appropriate and up-to-date databases.

  • Detection time: The system, although efficient in terms of accuracy, has a detection time of more than 500 s, which is much too high for a real-time application.

Recommendation: It is advisable to compare the proposed solution with other work on real-time detection for power monitoring systems, and explore optimization axes to reduce processing time.

Author Response

​​Comments 1:​​
The article does not provide a detailed architecture for full system implementation in a power monitoring environment. It focuses mainly on the methodology of detecting malicious traffic (via a stacking approach) and experimental validation with the NSL-KDD dataset. Although the document mentions the importance of safety in power monitoring systems and some requirements (such as the need for real-time processing), it does not describe in detail the full technical architecture or integration required to deploy this system into a power monitoring system.

Response 1:

Thank you for pointing this out . We agree with this comment. The power monitoring system generally connects the flow monitoring bypass. Our paper focuses on capturing, parsing, and storing packets, without the need for coupling with the power system. We have provided an explanation for this in the analysis section of Table 8 of the paper.

Comments 2:

Thank you for pointing this out . We agree with this comment. The Choice of dataset: The NSL-KDD database is not specifically designed for power monitoring applications as it dates back to 2009, whereas more recent databases exist (e.g. NetResec Public PCAPs (2021) or CIC DoH 2020 (2020)). In addition, this dataset does not specifically represent attacks targeting power monitoring systems.

Response 2:

Although the NSL-KDD dataset is not specifically designed for power monitoring systems, its attack categories (such as DoS, R2L) are similar to common threats in power grid network security (such as denial of service attacks against SCADA systems, illegal control instruction injection). Although there are updated datasets such as CIC IDS 2023, NSL-KDD remains a widely recognized benchmark for evaluating the generalization ability of anomaly detection models. We provided an explanation in section 3.1 of the paper.

Comments 3:

Detection time: The system, although efficient in terms of accuracy, has a detection time of more than 500 s, which is much too high for a real-time application.

Response 3:

Thank you for pointing this out . We agree with this comment. Real time performance is not the focus of our paper, as the flow detection bypass is usually connected to power monitoring, which can be detected afterwards. The ability to discover afterwards is more important. We have provided an explanation for this in the analysis section of Table 8 of the paper.